# A Network of MicroRNAs and mRNAs Involved in Melanosome Maturation and Trafficking Defines the Lower Response of Pigmentable Melanoma Cells to Targeted Therapy

**DOI:** 10.3390/cancers15030894

**Published:** 2023-01-31

**Authors:** Marianna Vitiello, Alberto Mercatanti, Maurizio Salvatore Podda, Caterina Baldanzi, Antonella Prantera, Samanta Sarti, Milena Rizzo, Alessandra Salvetti, Federica Conte, Giulia Fiscon, Paola Paci, Laura Poliseno

**Affiliations:** 1Institute of Clinical Physiology (IFC), National Research Council (CNR), 56124 Pisa, Italy; 2Oncogenomics Unit, Core Research Laboratory (CRL), ISPRO, 56124 Pisa, Italy; 3University of Siena, 53100 Siena, Italy; 4Unit of Experimental Biology and Genetics, Department of Clinical and Experimental Medicine, University of Pisa, 56126 Pisa, Italy; 5Institute for Systems Analysis and Computer Science “A. Ruberti” (IASI), National Research Council (CNR), 00185 Rome, Italy; 6Department of Computer, Control, and Management Engineering “A. Ruberti” (DIAG), Sapienza University of Rome, 00185 Rome, Italy

**Keywords:** melanoma, BRAF inhibitors, drug resistance, pigmentation, miRNAs, mRNAs, melanosome maturation, melanosome trafficking

## Abstract

**Simple Summary:**

Selective inhibitors of mutant BRAFV600E (BRAFi) have revolutionized the treatment of metastatic melanoma patients and represent a powerful example of the efficacy of targeted therapy. However, one of the main limitations of BRAFi is that treated cells put in place several adaptive response mechanisms, which initially confer drug tolerance and later provide a gateway for the insurgence of genetically acquired resistance mechanisms. We previously discovered that pigmentation is one of these adaptive response mechanisms. Upon BRAFi treatment, those cells that increase their pigmentation level are more resistant to BRAFi than those that do not. Here, we demonstrate that pigmentation limits BRAFi activity through an increase in the number of intracellular mature melanosomes. We also show that this increase derives from increased maturation and/or trafficking. In addition, we identify the miRNAs and mRNAs that are involved in these biological processes. Finally, we provide the rationale for testing a new combinatorial therapeutic strategy that aims at increasing BRAFi efficacy by blocking the adaptive responses that they elicit. This strategy is based on the combined use of BRAFi with inhibitors of pigmentation, specifically inhibitors of melanosome maturation and/or trafficking.

**Abstract:**

Background: The ability to increase their degree of pigmentation is an adaptive response that confers pigmentable melanoma cells higher resistance to BRAF inhibitors (BRAFi) compared to non-pigmentable melanoma cells. Methods: Here, we compared the miRNome and the transcriptome profile of pigmentable 501Mel and SK-Mel-5 melanoma cells vs. non-pigmentable A375 melanoma cells, following treatment with the BRAFi vemurafenib (vem). In depth bioinformatic analyses (clusterProfiler, WGCNA and SWIMmeR) allowed us to identify the miRNAs, mRNAs and biological processes (BPs) that specifically characterize the response of pigmentable melanoma cells to the drug. Such BPs were studied using appropriate assays in vitro and in vivo (xenograft in zebrafish embryos). Results: Upon vem treatment, miR-192-5p, miR-211-5p, miR-374a-5p, miR-486-5p, miR-582-5p, miR-1260a and miR-7977, as well as *GPR143*, *OCA2*, *RAB27A*, *RAB32* and *TYRP1* mRNAs, are differentially expressed only in pigmentable cells. These miRNAs and mRNAs belong to BPs related to pigmentation, specifically melanosome maturation and trafficking. In fact, an increase in the number of intracellular melanosomes—due to increased maturation and/or trafficking—confers resistance to vem. Conclusion: We demonstrated that the ability of pigmentable cells to increase the number of intracellular melanosomes fully accounts for their higher resistance to vem compared to non-pigmentable cells. In addition, we identified a network of miRNAs and mRNAs that are involved in melanosome maturation and/or trafficking. Finally, we provide the rationale for testing BRAFi in combination with inhibitors of these biological processes, so that pigmentable melanoma cells can be turned into more sensitive non-pigmentable cells.

## 1. Introduction

In metastatic melanoma, resistance to BRAF inhibitors (BRAFi) and MEK inhibitors (MEKi) remains one of the main drawbacks of targeted therapy [1]. While much research effort over recent years has been focused on understanding genetic resistance mechanisms that emerge after prolonged drug exposure, growing evidence indicates that adaptive mechanisms conferring early drug tolerance allow treated cells to survive and eventually develop genetic resistance. Targeting such adaptive mechanisms may thus contribute to preventing, rather than reversing, drug resistance [2,3].

Adaptive mechanisms coexist and include: enrichment of slow cycling JARID1B^+^ or drug efflux-prone ABCB5^+^ melanoma stem cells [4,5]; acquisition of a stress signature [6]; dedifferentiation toward a neural crest-like phenotype [6,7,8,9]; acquisition of an invasive MITF-low signature [6]; and acquisition of a MITF-high signature [10,11], resulting in increased pigmentation [6,12,13] and enhanced oxidative phosphorylation [14,15]. In turn, targeting each of these adaptive mechanisms has been shown to strongly potentiate the efficacy of BRAFi and/or MEKi and to strongly delay disease progression [9,16].

Within this framework, in recent years we have reported that not all metastatic melanoma cell lines display an increase in pigmentation upon treatment with BRAFi/MEKi. Some become more pigmented (pigmentable), while others do not (non-pigmentable). In addition, using the BRAFi vemurafenib (vem), we found that a visually evident increase in pigmentation is associated with an increase in the number of melanosomes, especially those that are fully mature (stage IV). We also discovered that vem-induced pigmentation is mediated by miR-211-5p (miR-211 for brevity). This microRNA (miRNA) is under the transcriptional control of MITF, which is the master transcriptional regulator of most of the genes belonging to melanin biosynthetic pathway (e.g., *PMEL*/*GP100*, *MLANA/MART1* [17], *GPR143/OA1* [18], *TYR*, *TYRP1*, *DCT* [19], miR-211 host gene *TRPM1* [20], *SLC45A2* [21], *UVRAG* [22], *SLC7A5* [23], and *RAB27A* [24]). MITF is de-repressed/re-activated by vem-induced blockage of the ERK pathway and upregulates all the above-mentioned genes. In turn, upregulated miR-211 targets EDEM1 and limits the degradation of TYROSINASE (TYR, the rate-limiting enzyme for melanin biosynthesis) through the ER-associated degradation (ERAD) pathway [25,26].

Crucially, we demonstrated that pigmentation is an adaptive cellular response that limits vem efficacy early on during treatment. Pigmentable lines in vitro, as well as metastatic melanoma patients with melanotic tumors, are characterized by a higher degree of resistance to BRAFi and/or MEKi compared to non-pigmentable lines and metastatic melanoma patients with amelanotic tumors. Furthermore, in both in vitro and xenograft experiments in zebrafish, the anti-proliferative effect of vem is counteracted by pigmentation inducers (e.g., positive modulators of TYR expression, such as miR-211 overexpression and kifunensine, which inhibit the expression and the enzymatic activity of EDEM1, respectively). Conversely, vem efficacy is potentiated by pigmentation inhibitors (e.g., negative regulators of TYR expression: LNA inhibitor of miR-211; inhibitors of TYR catalytic activity, such as phenylthiourea (PTU)) [25,26]. These latter experiments highlight the necessity of better targeting pigmentable melanoma cells, to fully unleash the efficacy of BRAFi/MEKi. In turn, they prompt us to dig further into the miRNome and transcriptome of pigmentable vs. non-pigmentable cell lines to identify, in an unbiased way, the full spectrum of biological pathways that characterize pigmentable cells, so that more informed therapeutic strategies can be devised to target them.

Here, we show that melanosome maturation and trafficking are the only biological processes that univocally distinguish the response of pigmentable 501Mel and SK-Mel-5 cells vs. non-pigmentable A375 cells to vem. We also pinpoint the small network composed of seven miRNAs and five mRNAs that provides the molecular bases for such biological processes. Finally, we show how such processes can be exploited therapeutically.

## 2. Materials and Methods

### 2.1. Cell Culture

Cells were grown at 37 °C in a humidified atmosphere with 5% CO_2_. A375, MNT1, SK-Mel-5 and 501Mel melanoma cell lines were cultured in DMEM High-Glucose medium (Euroclone, Milan, Italy) supplemented with 10% fetal bovine serum (Euroclone, Milan, Italy), 1% glutamine (Sigma-Aldrich, St. Louis, MO, USA) and 1% penicillin/streptomycin (Euroclone, Milan, Italy). Colo800 were cultured in RPMI medium (Euroclone, Milan, Italy) supplemented with 10% fetal bovine serum (Euroclone, Milan, Italy), 1% glutamine (Sigma-Aldrich, St. Louis, MO, USA) and 1% penicillin/streptomycin (Euroclone, Milan, Italy). The identity of each cell line was confirmed by fingerprinting, as reported in [27].

### 2.2. Drugs

Vemurafenib (vem, PLX-4032, #S1267) was purchased from Selleckchem, Houston, TX, USA.

Heparin (heparin sodium salt from porcine intestinal mucosa, #4784), Geneticin/G418 (G418 disulfate salt, #A1720) and puromycin (puromycin dihydrochloride, #P8833) were purchased from Sigma-Aldrich, St. Louis, MO, USA.

All drugs were diluted according to the manufacturer’s instructions.

### 2.3. Oligos

PCR and qRT-PCR oligos, siRNAs for mRNA knock-down or microRNA mimicking (mimics) were purchased from Eurofins Genomics, Ebersberg, Germany. LNA inhibitors were purchased from Qiagen, Hilden, Germany. For sequences, see Table 1.

### 2.4. Generation of 501Mel Cells That Stably Express TYR-mCherry Fluorescent Protein

The TYR-mCherry plasmid (kind gift of Dr. Carmit Levy, Tel Aviv University) was first linearized with BglII and then transfected in 501Mel cells, using Lipofectamine^TM^ 2000 (Thermo Fisher Scientific, Waltham, MA, USA). Transfected cells were selected using 1 mg/mL gentamicin. The G418-resistant population, which expressed the plasmid, was later enriched in mCherry-positive cells, using fluorescence activated cell sorting (FACSjazz, Becton Dickinson, Franklin Lakes, NJ, USA).

### 2.5. Transfection of miRNA Mimics and LNAs

Melanoma cells were seeded in 6-well plates to reach 80–90% confluency the day after. Seeded cells were transfected with Lipofectamine^TM^ 2000 (Thermo Fisher Scientific, Waltham, MA, USA) and 60 nM LNAs in DMEM High Glucose (Euroclone, Milan, Italy). Six hours post-transfection, the medium was changed to complete medium and cells were processed according to the protocol of downstream assays.

### 2.6. Melanin Content Evaluation

A total of 3 × 10^5^ melanoma cells were seeded in 60 mm plates and 24 h later, they were treated with either DMSO or the drug (vemurafenib). After 72 h of treatment, the cells were harvested and counted. Finally, pictures were taken on equal numbers of pelleted cells.

### 2.7. Transmission Electron Microscope Analysis

To evaluate morphological features, A375 and 501 Mel cells were seeded at 5 × 10^5^ cells/P100. The following day, they were treated with either DMSO or 2 µM vemurafenib for 72 h. At the end of this period, cells were harvested and pelleted by centrifugation. Pellets were washed three times with phosphate buffered saline (PBS) solution and fixed in 3% glutaraldehyde solution in 0.1 M cacodylate buffer, pH 7.2, for 2 h at 4 °C. Cells were then scraped off and post-fixed in 1% osmium tetroxide in 0.1 M cacodylate buffer for 2 h at room temperature. After rapid dehydration in a graded series of ethanol and propylene oxide, cells were embedded in an “Epon-Araldite” mixture. Ultrathin sections, obtained by a diamond knife on an Ultracut Reichert-Jung ultramicrotome, were placed on Formvar-carbon coated nickel grids, stained with uranyl acetate and lead citrate, and observed with a Jeol 100 SX transmission electron microscope. The quantification of the number of melanosomes per cell was performed by counting the number of melanosomes per unit of cytoplasmic area.

### 2.8. Quantification of Released Melanosomes

TYR-mCherry 501Mel cells were seeded in 100 mm plates (1 plate per experimental condition) and 24 h later they were treated with 5 µM vem or with DMSO vehicle. After 10 days, cells were harvested and counted to normalize the number of released melanosomes. Meanwhile, the supernatant was harvested and centrifuged at 7197× *g* for 30 min at 4 °C to remove debris and the cells remaining in the suspension. Then, to isolate melanosomes, it was subjected to ultracentrifugation at 27,000× *g* for 1 h at 4 °C. Finally, red melanosomes present in the pellet were quantified using flow cytometry (FACSjazz, Becton Dickinson, Franklin Lakes, NJ, USA).

### 2.9. Clonogenicity Assay

Cells were seeded (2 × 10^2^) in 6 cm plates in triplicate and treated appropriately. After 8 days, cells were fixed and stained with a 0.1% crystal violet, 4% formaldehyde solution. The number of colonies was normalized on the number of colonies obtained in the negative control. The average colony number ± SEM for each group was then used to create a bar graph.

### 2.10. Co-Colture Assay

Cells were seeded in the insert of a 6-well dish (1 µm pore size, #353102, Falcon®, upper chamber), and directly in the same 6-well dish (lower chamber).

The cell lines seeded in the lower chamber were Colo800, 501Mel, SK-Mel-5 or MNT1 cells that stably expressed EGFP because they were infected with pGIPZ-tGFP lentiviral vector [25], as well as 501Mel cells stably infected with pGIPZ-miR-211/SCR, hence over-expressing miR-211/SCR, and expressing EGFP as well [25].

The cell lines seeded in the upper chamber were TYR-mCherry 501Mel cells; TYR-mCherry and miR-211/SCR 501Mel cells.

The next day, cells were treated with a drug or with DMSO vehicle. The percentage of red cells in the lower chamber was evaluated 1 week after drug treatment using flow cytometry (FACSjazz, Becton Dickinson, Franklin Lakes, NJ, USA). For each sample, 10^4^ events were analyzed.

### 2.11. Xenograft in Zebrafish Embryos

TYR-mCherry 501Mel cells or TYR-mCherry miR-211-5p/SCR 501Mel cells were treated with the appropriate drug or with vehicle (DMSO) for 48 h. Cells were then harvested and injected into the yolk sac of 48 h post fertilization (hpf) zebrafish embryos, as reported in [25].

### 2.12. RNA Extraction and Quantification

For general purposes RNA was extracted using QIAzol reagent (Qiagen, Hilden, Germany), following the manufacturer’s instructions. For microRNA sequencing, RNA was extracted using miRNEasy Mini Kit (Qiagen, Hilden, Germany), following the manufacturer’s instructions. RNA was subsequently quantified using Nanodrop Lite (Thermo Fisher Scientific, Waltham, MA, USA).

### 2.13. DNAse Treatment and Retrotranscription

When analyzing mRNA expression, 1 µg of RNA was retrotranscribed using QuantiTect Reverse Transcription Kit (Qiagen, Hilden, Germany), following the manufacturer’s instructions and using an S1000 Thermal Cycler (Bio-Rad, Hercules, CA, USA).

The absence of genomic contamination of RNA was confirmed by performing a PCR reaction on the cDNA using PCR Master Mix (Thermo Fisher Scientific, Waltham, MA, USA) and the *ATPA1* primers [25]. These primers produce a genomic-derived amplicon of 300 bp and a cDNA-derived amplicon of 180 bp, allowing for genomic DNA contamination detection.

When analyzing miRNA expression, 250 ng of RNA was retrotranscribed using miScript II RT Kit (Qiagen, Hilden, Germany), following the manufacturer’s instructions and using an S1000 Thermal Cycler (Bio-Rad, Hercules, CA, USA).

### 2.14. Real-Time PCR

Quantitative PCR was performed with SsoAdvanced Universal Supermix (Bio-Rad, Hercules, CA, USA) on a CFX96 Real-Time System (Bio-Rad, Hercules, CA, USA). A melting curve was performed after each PCR reaction to confirm the specificity of the primers. All reactions were performed in duplicate. Data were analyzed using CFX Manager Software (Bio-Rad, Hercules, CA, USA).

### 2.15. Protein Extraction and Western Blot Analysis

Melanoma cells (10^6^) were resuspended in 30 µL of lysis buffer (Tris HCl 50 mM, 1% TritonX100, 0.25% of NaDeoxicholate, PMSF 1 mM, Orthovanadate 2 mM, proteinase inhibitors cocktail). The mixture was incubated for 30 min on ice, then sonicated for 30 min, and finally centrifuged at 14,000× *g* rpm for 30 min at 4 °C. The supernatant was then quantified using Bradford reagent and read at 590 nm.

The samples were then heated at 95 °C for 5 min, separated on 10% SDS-polyacrylamide gels (Mini-PROTEAN Precast gel, Bio-Rad, Hercules, CA, USA) and electrotransferred to nitrocellulose membranes (Trans-Blot Turbo Midi 0.2 µm Nitrocellulose Transfer Packs, Bio-Rad, Hercules, CA, USA), using a Trans-Blot Turbo system (Bio-Rad, Hercules, CA, USA). Membranes were blocked at room temperature for 2 h using 3% milk in TBST, and then incubated overnight at 4 °C with the following primary antibodies:-anti-DCT (#sc-74439, Santa Cruz Biotechnology, Dallas, TX, USA; mouse monoclonal antibody, dilution 1:1000 in 3% milk in TBST);-anti-GAPDH (#2118, Cell Signaling, Danvers, MA, USA; rabbit polyclonal antibody, dilution 1:3000 in 3% milk in TBST);-anti-PMEL17 (#sc-377325, Santa Cruz Biotechnology, Dallas, TX, USA; mouse monoclonal antibody, dilution 1:1000 in 3% milk in TBST);-anti-TYR (#sc-20035, Santa Cruz Biotechnology, Dallas, TX, USA; mouse monoclonal antibody, dilution 1:1000 in 3% milk in TBST);-anti-TYRP1 (#sc-166857, Santa Cruz Biotechnology, Dallas, TX, USA; mouse monoclonal antibody, dilution 1:1000 in 3% milk in TBST).

The detection of primary antibodies was performed using alkaline phosphatase-conjugated secondary antibodies and enhanced chemiluminescence reagents (Clarity Western ECL Substrate, Bio-Rad, Hercules, CA, USA).

### 2.16. Statistical Analysis

Data were analyzed with unpaired t test (GraphPad Prism, GraphPad Software Inc.). Values of *p* < 0.05 were considered statistically significant (* *p* < 0.05, ** *p* < 0.01, *** *p* < 0.001, **** *p* < 0.0001). The mean ± SEM of three independent experiments were reported.

### 2.17. Deep Sequencing

#### 2.17.1. Sample Preparation

In total, 10^6^ 501Mel and SK-Mel-5 cells were seeded in 100mm plates, and 24 h later they were treated with DMSO or 2 µM vemurafenib for 48 h. Total RNA was then extracted using the miRNeasy Mini Kit (Qiagen, Hilden, Germany). To account for biological variability, three independent replicates of this experiment were subjected to sequencing.

#### 2.17.2. Library Generation and Sequencing

For smallRNA-seq, 1 µg of total RNA (including small RNA fraction) with RIN > 8 (2100 Bioanalyzer, Agilent technologies) was used for library preparation with a TruSeq smallRNA Sample Preparation kit (Illumina, San Diego, CA, USA) according to the manufacturer’s instructions. cDNA libraries were loaded at six-plex level of multiplexing (about 4 million reads per samples) into a flow cell V3 and sequenced in a single-reads mode (50 bp) on a MiSeq sequencer (Illumina, San Diego, CA, USA).

For RNA-seq, RNA libraries were prepared from 1 µg of the same total RNA used for smallRNA-seq, using the TruSeq stranded RNA Sample Preparation Kit (Illumina, San Diego, CA, USA), following the manufacturer’s suggestions. Libraries were prepared for sequencing and then sequenced on single-end 75 bp mode on NextSeq 500 (Illumina, San Diego, CA, USA), obtaining about 40 million reads per sample. In this case, both the library preparation and the sequencing were performed by IGA Technology Services, Udine, Italy.

#### 2.17.3. Primary Analysis, Clustering, and Differential Expression Analysis

For smallRNA-seq, the raw sequences produced by the MiSeq sequencer (Illumina, San Diego, CA, USA) were processed by FastQC, version 0.11.7, for quality check. The primary reads obtained were trimmed off adapter sequences using Cutadapt, version 1.9.1 [28]. The remaining high-quality reads, ranging between 17 and 35 bp after clipping, were clustered for unique hits and mapped to known human pre-miRNA sequences from mirBase, release 21, by the miRExpress tool, version 2.1.3 [29]. Finally, miRNA differential expression analysis was performed in the R environment using Bioconductor’s package DESeq2, version 1.36.0 [30].

For RNA-seq, the analysis of samples belonging to 501 Mel and SK-Mel-5 cell lines was performed by IGA Technology Services, Udine, Italy. The primary analysis was performed with Bcl2Fastq 2.0.2 for base calling and demultiplexing, Cutadapt v1.11 and ERNE software for adapter masking and trimming. The secondary analysis was employed with STAR software for alignments on the reference genome, with Stringtie and htseq-count for transcripts count and DESeq2 to perform comparisons between expression levels of genes and transcripts. For samples of the A375 cell line, we performed the differential expression analysis on the RNA-seq raw dataset retrieved from NCBI Gene Expression Omnibus, GEO accession GSE89127. We employed Bioconductor’s DESeq2 for differential expression analysis of this RNA-seq data as well.

For all seq data, variance stabilizing transformed (VST) count data were exploited to study the relationship between samples. For the hierarchical clustering, we used the Euclidean distance as a distance metric and completed the agglomeration method.

Furthermore, for all seq data, we considered as differentially expressed only those items with a minimum read count value of 32 (background noise threshold), a log2 fold change (log2FC) higher than +0.4 or lower than −0.4, and an adjusted *p*-value 0.05.

#### 2.18. clusterProfiler Analysis

The enriched functional profile analysis of SK-Mel-5, 501Mel and A375 RNA-seq was performed using the clusterProfiler package [31,32]. This package supports the functional characteristics of both coding and non-coding genomics data with up-to-date gene annotation. Datasets obtained from multiple treatments and time points can be analyzed and compared in a single run, easily revealing functional consensus and differences among distinct conditions.

According to the compareCluster function, the SK-Mel-5 cluster consisted of 4354 genes, the 501Mel cluster consisted of 5689 genes, and the A375 cluster consisted of 6408 genes. The list of genes belonging to one or more clusters was obtained through the above-mentioned DE analysis. The org.Hs.eg.db package was used for genome-wide annotation, mapping was obtained using Entrez IDs, and Gene Ontology: Biological Process was used as a reference database. A 0.05 *p*-value cut-off was applied. All of the packages were used through R, an open source programming environment.

### 2.19. WGCNA Analysis

#### 2.19.1. Construction of Weighted Gene Co-Expression Network and Identification of Hub Modules

Weighted correlation network analysis (WGCNA) [31] is a method for finding clusters (modules) of highly correlated genes and for relating the modules to external traits, by eigengene network methodology. This method is used to identify sets of genes that are expressed together in sample datasets, which can aid in the interpretation of the relationship between RNA-seq data and phenotype traits. The data were processed with WGCNA package version 1.70 for GNU R software environment, version 4.1.

First, analyzing scale-free topology for multiple hard thresholds, we selected the 0.85 threshold value needed for network construction. Moreover, correlation matrix of gene co-expression and dissimilarity coefficients of nodes were computed.

Then, genes with similar expression patterns were identified, clustered to define modules with different colors for visualization, and represented by hierarchical clustering tree.

Finally, the relationship between each module and the external traits was analyzed: DMSO vs. vemurafenib treatment, pigmentable vs. non-pigmentable cell lines, expression levels of the seven DEmiRs (hsa-miR-192-5p, hsa-miR-211-5p, hsa-miR-374a-5p, hsa-miR-486-5p, hsa-miR-582-5p, hsa-miR-1260a, and hsa-miR-7977). Correlation values are represented as a heatmap graph.

#### 2.19.2. clusterProfiler Analysis of the Genes Belonging to WGCNA Modules

To understand the biological meaning of the genes belonging to each module, the R package clusterProfiler Ver. 4.2.2 [32,33] was used to perform gene ontology enrichment analysis (*p*-value < 0.05 was regarded as significant).

### 2.20. SWIMmeR Analysis

SWIM (SWItchMiner) is a novel methodology that, starting from gene expression profiles of two conditions of interest, computes the differentially expressed genes within the co-expression network framework and combines this information with a structured network of correlated patterns to identify key genes likely associated with drastic changes in cell phenotypes. SWIM first builds a correlation network, where nodes are differentially expressed genes and a link occurs if their expression profiles are highly correlated or anti-correlated (according to a defined threshold). Then, it searches for modules in this network and assesses the functional roles of network nodes according to their ability to convey information within and between modules. By combing the topological properties with the inter- and intra-module connections, SWIM identifies a small pool of genes (known as switch genes) that are associated with intriguing patterns of molecular co-abundance and play a crucial role in the observed phenotype, especially in the transitions between one condition to another one [34]. The strength of SWIM is to emphasize the importance of negative regulation by explicitly considering, in addition to the right tail (i.e., positive correlation between gene pairs), also the left tail (i.e., negative correlation between gene pairs) of the correlation distribution, allowing the interpretation of negative edges within a complex network representation of functional connectivity.

SWIM methodology has been implemented in R (SWIMmeR, 2021) [35] and has been successfully applied to a broad range of phenotype-specific scenarios, spanning from complex diseases [34,36,37,38,39,40] to grapevine berry maturation [41].

#### 2.20.1. miRNAs-Module Association

To investigate the relationship between the modules identified by SWIM and the differentially expressed miRNAs, we correlated the miRNA expression profiles with the module eigengene (ME) of each module identified by SWIM analysis. The ME is defined as the first principal component of a given module and can be considered a representative of the gene expression profiles in that module. The sign of this correlation encodes whether the miRNA has a positive or a negative relationship with the ME of each module. If the correlation of a given miRNA with a given module is close to 0, that miRNA is not associated with that module. On the other hand, if the correlation is close to 1 or −1, the positive or negative association between the miRNA and the module is high.

#### 2.20.2. Functional Enrichment Analysis

The associations between switch genes and functional annotations such as KEGG pathways [42] and GO terms [43] were obtained by using the EnrichR web tool [44]. *p*-values were adjusted with the Benjamini–Hochberg method, and a threshold equal to 0.05 was set to identify functional annotations significantly enriched amongst the selected gene lists.

### 2.21. Gene Correlations Analysis

Gene correlation analyses were performed using the data explorer tool of the Dependency Map web portal, a database that allows one to identify genetic and pharmacologic dependencies (DepMap, Broad (2022): DepMap 22Q2 Public Figshare Dataset https://doi.org/10.6084/m9.figshare.19700056.v2 (accessed on August 2022)). The TPM gene expression values of the protein-coding genes were inferred from RNA-seq data of cell lines, using the RSEM tool, and were reported after log2 transformation, using a pseudo-count of 1; log2(TPM+1) [45]. Through the DepMap data explorer tool, the Pearson correlation coefficient of *RAB27A*, *RAB32*, *OCA2*, *TYRP1*, and *GPR143* was calculated against *TRPM1* and *TRPM3*.

## 3. Results and Discussion

### 3.1. Identification of Differentially Expressed miRNAs and mRNAs in Pigmentable vs. Non-Pigmentable Cell Lines upon Vem Treatment

The aim of this study was to discover the biological framework that explains the difference in sensitivity to vem displayed by pigmentable versus non-pigmentable melanoma cells. To achieve this goal, we looked for differentially expressed (DE) genes (both miRNAs (DEmiRs) and mRNAs (DEmRNAs)) between two pigmentable metastatic melanoma cell lines (501Mel and SK-Mel-5) and one non-pigmentable metastatic melanoma cell line (A375). As shown in Figure 1a–c, genes known to be involved in melanin synthesis are only expressed in 501Mel and SK-Mel-5 cells (see also Appendix A). Accordingly, melanosomes are only visible in the cytoplasm of 501Mel (Figure 1d,e) and SK-Mel-5 cells. Furthermore, A375 cells do not become pigmented upon vem treatment (Figure 1f) and are more sensitive to the drug compared to 501Mel and SK-Mel-5 cells, which do become pigmented. Specifically, vem IC50 is 0.04 µM ± 0.16 for A375 cells vs. 0.1 µM ± 0.06 for 501Mel cells and 0.13 µM ± 0.04 for SK-Mel-5 cells (see also [25]).

To identify DEmiRs and DEmRNAs, we treated 501Mel and SK-Mel-5 cells with 2 µM vem for 48 h, and then subjected them to smallRNA-seq and RNA-seq, respectively (Figure 1g). For each condition, 3 biological replicates were analyzed (Appendix A). NGS data of A375 cells were already available: the smallRNA-seq of cells treated with 2 µM vem for 48 h was previously published by us [25]. The RNA-seq of cells treated with 1 µM vem for 48 h was retrieved from GEO database ([46], GSE89129).

We considered as differentially expressed in vem vs. DMSO those miRNAs showing 32 reads on average, log2FC > 0.4 or log2FC < −0.4, and adjusted *p*-value < 0.05 (Appendix A). The bar graph in Figure 1h, left shows that the number of up- and down-regulated miRNAs is approximatively the same in all three cell lines. The Venn diagram in Figure 1h, right shows DE miRNAs that are cell line-specific (A375: 165; 501Mel: 24; SK-Mel-5: 9), DE miRNAs that are shared by all three cell lines (19), DE miRNAs that are shared between A375 and only one other pigmentable cell line (501Mel and A375: 22; SK-Mel-5 and A375: 26), and DE miRNAs that are specific to pigmentable cell lines, i.e., shared by 501Mel and SK-Mel-5 cells, but not by A375 cells. There are seven such DE miRNAs, which, from now on, we call DEmiRs for brevity, and they represent our group of interest: miR-192-5p, miR-211-5p, miR-374a-5p, miR-486-5p and miR-582-5p are up-regulated in vem compared to DMSO (UP DEmiRs), while miR-1260a and miR-7977 are down-regulated (DOWN DEmiRs) (Figure 1i,j).

Analogously, we considered as differentially expressed in vem vs. DMSO those mRNAs showing 32 reads on average, log2FC > 0.4 or log2FC < −0.4, and adjusted *p*-value < 0.05 (Appendix A). In 501Mel and in SK-Mel-5 cells, 5689 and 4354 DE RNAs were identified, respectively (Figure 1k and Appendix A). Among them, 2522 are differentially expressed in both cell lines, 2371 in a concordant way (1203 UP-UP and 1168 DOWN-DOWN). In A375 cells, 6408 DE RNAs were identified (Appendix A). After excluding the 2371 DE RNAs that are shared by 501Mel and SK-Mel-5 cells, as well as those that are differentially expressed in a concordant way in A375 cells, 1301 DE RNAs remained, which are specific to pigmentable melanoma cell lines (Figure 1l). This number reduces to 1275 DEmRNAs when the 26 RNAs not present in the HGNC database (https://www.genenames.org/ (accessed on 31 August 2022)) are excluded (Appendix A).

### 3.2. Identification of Biological Processes Selectively Enriched in Pigmentable vs. Non-Pigmentable Cell Lines upon Vem Treatment

To identify the transcriptional pathways that distinguish the response of pigmentable vs. non-pigmentable cells to vem, DEmRNAs were analyzed using three different methods: enriched functional profiles (clusterProfiler) analysis, WeiGhted Correlation Network Analysis (WGCNA) and SWIMmeR analysis.

#### 3.2.1. Enriched Functional Profiles Analysis by clusterProfiler

The enriched functional profiles (clusterProfiler) analysis, performed on DE RNAs (6408 for A375 cells, 5689 for 501Mel cells, and 4354 for SK-Mel-5 cells), identified a total of 1430 enriched and non-redundant GO: Biological Processes (BPs) (Appendix A). Narrowing down to the 200 top-scoring BPs (*p*-value < 0.05), we found that most of them can be grouped into 12 macroBPs, which are listed in Figure 2a and represented as pie charts in Appendix A. All three melanoma cell lines share macroBPs related to cell growth and cell cycle, cellular organization, nucleic acid and proteins related processes, neurogenesis, protein localization, response to ROS and O_2_ respiration, and cellular response to stimuli. This is expected after BRAFi treatment, as it reflects drug-induced cell cycle arrest [25] and a decrease in protein translation [47]. It is also consistent with adaptive cellular responses that melanoma cells use to limit BRAFi: metabolic rewiring from glycolysis to oxidative phosphorylation [14], acquisition of a stress signature [6], and dedifferentiation toward a neural crest-like phenotype [6,7,8,9]. Interestingly, the macroBP “pigmentation”, which is an additional adaptive response against BRAFi [6,12,13,25], is the only one that we found to be enriched in pigmentable 501Mel and SK-Mel-5 cell lines and not in non-pigmentable A375 cell line. The macroBP “pigmentation” is composed of seven BPs (GO:0046148-pigment biosynthetic process; GO:0042440-pigment metabolic process; GO:0048066-developmental pigmentation; GO:0006582-melanin metabolic process; GO:0042438-melanin biosynthetic process; GO:0030318-melanocyte differentiation; GO:0050931-pigment cell differentiation, Appendix A), for a total of 64 genes. Fifteen out of these 64 genes are also DEmRNAs: *ABCC2*, *ADA*, *CPOX*, *DDT*, *GPR143*, *MREG*, *OCA2*, *PMEL*, *PPARGC1A*, *RAB27A*, *RAB32*, *SLC45A2*, *TRPC1*, *TYR*, and *TYRP1* (Figure 2b).

#### 3.2.2. Modules Analysis by WGCNA

The weighted correlation network analysis (WGCNA) allowed to find modules (clusters of densely interconnected, highly correlated genes) using weighted data (i.e., mRNA expression levels) as input information. It also allowed for correlating such modules with variables (in our case: treatment (DMSO vs. vem), cellular properties (pigmentable vs. non-pigmentable cell lines), and expression level of DEmiRs). Therefore, it was suitable to define the network of mRNAs and miRNAs involved in vem response of pigmentable vs. non-pigmentable cell lines.

Gene modules were identified using unsupervised clustering, without a priori defined gene sets [31]. The Dynamic Tree Cut algorithm considered 14634 RNAs (total number of RNAs expressed in A375, 501Mel and SK-Mel-5 cells) and grouped them into 12 modules, plus one more that collected all the leftovers and is shown in grey (Figure 2c and Appendix A). A hierarchical clustering dendrogram of modules, labeled by their colors, is presented in Appendix A. Modules contained a variable number of genes, ranging from 45 to 9178 (Appendix A).

Information about treatment (DMSO/vem), cellular properties (pigmentable/non-pigmentable) and expression level of DEmiRs were then used to obtain a co-expression analysis of the 12 modules. This analysis confirmed that the gene expression profile of A375 cells is different compared to that of the other two cell lines. Conversely, 501Mel and SK-Mel-5 maintain their identity but cluster together (Appendix A). In addition, the co-expression analysis allowed us to highlight three modules with specific features: *chocolate4*, *darkturquoise*, and *firebrick4* are characterized by discordant correlation in pigmentable vs. non-pigmentable cell lines, as well as in UP DEmiRs (miR-192-5p, miR-211-5p, miR-374a-5p, miR-486-5p, miR-582-5p) vs. DOWN DEmiRs (miR-1260a and miR-7977) (Figure 2d). Among the genes that compose *chocolate4*, *darkturquoise* and *firebrick4* modules, 160 out of 824 (19.4%), 72 out of 912 (7.9%) and 472 out of 9178 (5.1%) are DEmRNAs, respectively (Appendix A). Focusing on *chocolate4* as the module with the highest percentage of DEmRNAs, we performed clusterProfiler analysis and we noticed that, in accordance with the analysis described above, it contains four pigmentation-related GOs (GO:0048066-developmental pigmentation, GO:0006582-melanin metabolic process, GO:0030318-melanocyte differentiation, GO:0042438-melanin biosynthetic process, Appendix A), for a total of eight genes. Four of them are DEmRNAs: *OCA2*, *RAB27A*, *RAB32*, and *TYRP1* (Figure 2e).

#### 3.2.3. Switch Genes Analysis by SWIMmeR

We performed SWIMmeR analysis with the aim being to “cross-validate” and reinforce WGCNA data, as well as to overcome its limitations, since it cannot discriminate in the direction (positive or negative) of gene correlation [39].

SWIMmeR analysis was performed on the 2371 RNAs that are differentially expressed in a concordant way in 501Mel and SK-Mel-5 cells (Figure 1k). In this analysis, we identified three modules: module #1 (M1), which mainly contains switch genes, and modules #2 and #3 (M2, M3), which mainly contain the negative nearest neighbors of switch genes, i.e., genes that show a strong negative correlation with switch genes (Figure 2f, left column and Appendix A). We then evaluated the correlation between the three modules and the seven DEmiRs. As reported in Figure 2f, second to last column, UP DEmiRs show a positive correlation with switch genes (module #1) and a negative correlation with the negative nearest neighbors of switch genes (modules #2 and #3). Conversely, DOWN DEmiRs show a positive correlation with module #2. These data suggest that switch genes are directly regulated by vem treatment and go hand in hand with the five upregulated DEmiRs.

From the 363 switch genes that compose module #1, we excluded 37 that are also switch genes in non-pigmentable A375 cells (column S in Appendix A). The remaining 326 pigmentable cell-specific switch genes were analyzed for enriched GO:BPs. There are five BPs that better represent the two pigmentable cell lines under vem treatment (adjusted *p*-value < 0.05). Consistently with the output of clusterProfiler and WGCNA, four of them are related to pigmentation: GO:0032401-establishment of melanosome localization, GO:0032402-melanosome transport, GO:0051904-pigment granule transport, GO:0032400-melanosome localization (Figure 2g). Interestingly, the fifth BP is related to lipid metabolism (GO:0019395-fatty acid oxidation), and pigmented melanomas have been recently shown to rely on fatty acids as a fuel source for oxidative phosphorylation (https://doi.org/10.1101/2022.05.04.490656 (accessed on 31 August 2022)). Therefore, this BP is still related to pigmentation, although indirectly.

The four BPs directly related to pigmentation are all composed of the same five genes, which are DEmRNAs as well (Figure 2h): *BBS5*, *GPR143*, *MLPH*, *MYO5A*, and *RAB27A*.

We decided to prioritize for further analysis five DEmRNAs that are selected by at least two out of the three analytical approaches used, namely: *GPR143*, *OCA2*, *RAB27A*, *RAB32*, and *TYRP1* (Figure 2i).

### 3.3. Validation of DEmiRs and DEmRNAs Identified by Bioinformatic Analyses

The integrated analysis of differentially expressed miRNAs and mRNAs in pigmentable vs. non-pigmentable melanoma cell lines upon vem treatment allowed us to define a short list of seven DEmiRs and five DEmRNAs, which we attempted to validate in independent datasets.

Our smallRNA-seq analysis of melanoma cell lines treated with vem is a unicum, with no analogous miRNomes present in common databases. Conversely, the datasets reporting RNA-seq analyses of melanoma cell lines treated with vem are several and were interrogated to corroborate our findings. All five DEmRNAs are upregulated upon vem treatment in both 501Mel and SK-Mel-5 cells, but not in A375 cells (Figure 3a, upper). This upregulation was confirmed in independent datasets of 501Mel cells (GSE104869) and of A375 cells (GSE190071) (Figure 3a, middle). It was also confirmed in additional pigmentable cell lines: UACC62 (GSE64741) and Colo800 (GSE64741) (Figure 3a, lower and Appendix A).

Next, we looked into the roles that the seven DEmiRs and the five DEmRNAs play in melanoma. Unfortunately, few data are available about six out of the seven DEmiRs. Specifically, there are no published data on miR-486-5p and miR-7977. There is only one publication on miR-374a-5p [48] and one on miR-582-5p [49], both pointing towards an oncosuppressive role. The publications on miR-192-5p are contradictory: one describes it as a tumor suppressor miRNA that serves as an effector of metformin-induced growth and motility suppression in melanoma cells [50], while another describes it as an oncogenic miRNA that is induced in hypoxic melanoma cells, is transferred into adjacent cytotoxic T lymphocytes through a gap junction-dependent mechanism, and contributes to suppressing their antitumoral activity [51]. Similarly, there is one publication that implies miR-1260a as an oncosuppressive miRNA whose plasma levels increase upon surgical resection of metastatic melanoma [52]. However, we identified miR-1260a-5p as an oncogenic microRNA that sustains *BRAF* expression at the translational level [53].

Within this landscape of fragmentary knowledge, miR-211 stands out as a well established oncogenic miRNA [25,54,55,56,57] that is associated with worse prognosis in metastatic melanoma (Figure 3b, see also [25,58]). miR-211 exerts its oncogenic functions both in non-cell-autonomous and in cell-autonomous ways: secreted through melanosomes, it contributes to reprogram fibroblasts into cancer associated fibroblasts (CAFs), hence promoting the formation of the dermal tumor niche [55]. Being under the transcriptional control of MITF, miR-211 is upregulated upon vem treatment and confers drug resistance by means of two main mechanisms: it rewires metabolism toward mitochondrial respiration [56], and as we demonstrated in our previous work, it sustains the expression of TYROSINASE enzyme, ultimately leading to an increase in the number of mature stage IV melanosomes [25,26,59]. The pro-pigmentation activity of miR-211, which we confirm here as well (Figure 3c), renders this DEmiR a positive control for our analytical approach.

The involvement of all five DEmRNAs in pigmentation-related processes is supported by the fact that on one side they show a positive correlation with *TRPM1* (host gene of miR-211 and marker of pigmentable melanoma cells), while on the other they show no correlation or even anticorrelation with *TRPM3* (host gene of miR-204-5p and marker of non-pigmentable melanoma cells) (Appendix A, see also [25]). Furthermore, according to GEPIA (http://gepia.cancer-pku.cn/ (accessed on 31 August 2022)), they are all associated with worse prognosis in metastatic melanoma (Figure 3d, see also [54,60,61]), with an oncogenic role already established for *GPR143*, *OCA2*, *RAB27A* and *TYRP1*.

GPR143/OA1 is a melanosome G-protein-coupled receptor, whose mutations have been associated with type I ocular albinism [62]. GPR143 protein sustains MITF expression, and hence PMEL expression, therefore contributing to correct melanosome maturation [63]. In melanoma, GPR143 overexpression has been shown to promote cell migration [64].

OCA2 (Oculocutaneous Albinism type 2) is a melanocyte-specific transporter protein, whose mutations have been associated with type II oculocutaneous albinism. OCA2 protein is supposed to be an integral membrane protein involved in the transport of tyrosine, a precursor of melanin. In melanoma, *OCA2* is under the transcriptional control of the oncogenic cAMP-regulated transcription co-activators 1–3 (CRTC1–3) and serves as an effector of their pro-survival and pro-motility functions [65].

TYROSINASE RELATED PROTEIN 1 (TYRP1) is a melanosomal enzyme that belongs to the tyrosinase family and is involved in the biosynthetic pathway of eumelanin. Mutations in TYRP1 are in fact associated with type III oculocutaneous albinism. However, it remains to be established whether human TYRP1 is endowed with catalytic activity like mouse Tyrp1 (it catalyzes the oxidation of 5,6-dihydroxyindole-2-carboxylic acid (DHICA) into indole-5,6-quinone-2-carboxylic acid (IQCA)) or whetehr it acts as a chaperone of other enzymes belonging to eumelanin biosynthetic pathway [66]. Interestingly, the oncogenic role mainly comes from *TYRP1* mRNA, rather than TYRP1 protein. This is because *TYRP1* mRNA sequesters the oncosuppressive miR-16, leading to the de-repression of oncogenic RAB17 [67].

The protein encoded by *RAB32* is a member of the RAB family of small GTPases. Its role in melanoma has not been studied yet, but it is known to be involved in endosomal trafficking, namely in the sorting of cargos necessary for melanin synthesis [68]: together with RAB38, RAB32 carries TYR, TYRP1 and DCT/TYRP2 enzymes from early/recycling endosomes to early-stage melanosomes, allowing their maturation [69]. *Rab32/Rab38* double knockout mice show coat and eye pigment dilution [70].

The protein encoded by *RAB27A* is a member of the RAB family of small GTPases as well, but it is involved in melanosome transport rather than biogenesis. Mature stage IV melanosomes reach the tips of melanocyte dendrites traveling along microtubules, and then the protein complex composed of RAB27A, Melanophilin and Myosin Va links them to actin filaments, so that they can be transferred to keratinocytes [70,71,72]. RAB27A is oncogenic in melanoma: amplified and/or overexpressed, it has been shown to promote proliferation and motility/metastatization [73]. However, at the mechanistic level, these biological activities have been mostly attributed to its function as a mediator of exosome release, rather than melanosome transfer [74]. Mutations in RAB27A are responsible for type 2 Griscelli syndrome (GS2), which has albinism among its traits [70].

### 3.4. Validation of the Pigmentation-Related Biological Processes Identified by Bioinformatic Analyses

As best highlighted by SWIMmeR analysis, UP DEmiRs and DEmRNAs are positively, rather than negatively, correlated (see Figure 2f). Interestingly, *TRPM1*/miR-211-5p UP DEmiR [20] and all five DEmRNAs (*GPR143* [18], *RAB27A* [24], *TYRP1* [19], *OCA2*, and *RAB32* (https://maayanlab.cloud/Harmonizome/gene_set/MITF-21258399-MELANOMA-HUMAN/CHEA (accessed on 31 August 2022) + Transcription + Factor + Binding + Site + Profiles)) are transcriptional targets of MITF, the master regulator of the melanocytic lineage. MITF is de-repressed/re-activated due to vem-induced blockage of the ERK pathway [14], and evidently superimposes its signature on the miRNome and the transcriptome of pigmentable melanoma cells. In light of this consideration, we hypothesize that, although direct interactions among DEmiRs and DEmRNAs have already been validated (https://mirtarbase.cuhk.edu.cn/~miRTarBase/miRTarBase_2022/php/index.php (accessed on March 2022)), the relevant targets of UP DEmiRs in the context of vem treatment should be searched for within module #2 or #3, rather than within module #1. We also decided to focus on the analysis of the biological processes in which DEmiRs and DEmRNAs are involved as a whole, before undertaking the study of specific functions and one-to-one interactions.

The pigmentation-related GO:BPs that emerged can be divided into two macro groups: clusterProfiler and WGCNA analysis identified BPs revolving around melanin synthesis (hence melanosome maturation to stage IV), while SWIMmeR analysis uniquely identified BPs revolving around melanosome transport. Specifically, OCA2, RAB32, TYRP1 and miR-211 are involved in melanin synthesis, while GPR143 and RAB27A are also involved in melanosome transport (Figure 3e,f). Since a vem-induced increase in the number of mature stage IV melanosomes was previously demonstrated by us [25], we checked whether melanosome trafficking is potentiated as well. In Figure 3g–i, we take advantage of 501Mel cells that stably express TYR-mCherry (Figure 3g) to show that, upon vem treatment, there is in fact an increase in the release (Figure 3h) and in the uptake (Figure 3i) of red fluorescent melanosomes. In turn, such an increase in overall melanosome trafficking prompted us to rule out the possibility that melanosomes impair vem activity by exporting the drug outside rather than by acting inside the cells [75,76]. To address this point, we used ultracentrifugation to isolate red melanosomes from the supernatant of TYR-mCherry 501Mel cells. After counting, melanosomes were administered to naïve 501Mel cells that were subsequently treated with vem. We observed that cells receiving extra melanosomes become more resistant to vem (Figure 3j). This result strongly suggests that it is in fact the increase in the number of intracellular melanosomes that confers resistance to vem, rather than their increased mobility. Nevertheless, this result also unveils that, besides the use of drugs that impair melanin synthesis/melanosome maturation (e.g., the TYROSINASE inhibitor phenylthiourea (PTU) [25]), an effective strategy to prevent vem-induced pigmentation could be the use of drugs that decrease the number of intracellular melanosomes by favoring their release and/or preventing their uptake.

Melanosome maturation and their transport to dendrite tips have been studied in detail. Conversely, the actual transfer of mature melanosomes from melanocytes to keratinocytes remains a conundrum. The proposed mechanisms are as follows: cytophagocytosis; coupling of exocytosis with PAR-2-dependent phagocytosis; shedding in vescicles; and membrane phusion [72]. The transfer of mature melanosomes from one melanoma cell into another is even less well known: no mechanistic models have been proposed yet, let alone inhibitors developed. Assuming that the mechanism is the same, in the absence of drugs that favor melanosome release, we tested heparin (Hepa, a known inhibitor of melanosome uptake [77]), expecting a decrease in resistance to vem. Indeed, we found that vem-induced increase in melanosome uptake is hampered (Appendix A). Consistently, we also found that, when 501Mel cells pretreated with vem + Hepa are xenografted in zebrafish embryos, they show an increase in vem sensitivity compared to 501Mel cells pretreated with vem alone (Appendix A).

Having demonstrated that mature melanosomes confer resistance to vem from the inside of melanoma cells, we have in place the right experimental setting for the definition of the role played by each DEmiR and DEmRNA. For example, 501Mel cells stably overexpressing miR-211, which become more pigmented and are more resistant to vem compared to control cells [25], show neither an increase in release nor in uptake of melanosomes (Appendix A), suggesting that the contribution of individual DEmiR/DEmRNA can be confined to only maturation or only trafficking. Conversely, it is plausible that there are genes involved in both, as attested by the GO:BPs in which RAB27A is listed (Figure 3e). In this respect, it will be interesting to evaluate whether the inhibition of RAB27A ultimately produces an increase or a decrease in intracellular mature melanosomes, and, consequently, whether resistance to vem increases or decreases. A lot will likely depend on the mechanism of inhibition, since a selective decrease in its pro-maturation activity is desirable, while a combined decrease in its transport activity (e.g., through the inhibition of its interaction with melanophilin [78,79]) would result in a detrimental accumulation of mature melanosomes that fail to be released. Finally, it will be interesting to test whether or not RAB27A inhibitors that block exosome release have an impact on pigmentation and vem resistance as well [80].

## 4. Conclusions

We performed deep sequencing of two pigmentable cell lines (501Mel and SK-Mel-5) vs. one non-pigmentable cell line (A375) before and after vem treatment, and we identified seven DEmiRs (miR-192-5p, miR-211-5p, miR-374a-5p, miR-486-5p, miR-582-5p, miR-1260a, and miR-7977) and five DEmRNAs (*GPR143*, *OCA2*, *RAB27A*, RAB32, and *TYRP1*). DEmiRs and DEmRNAs belong to GO:BPs related to pigmentation. This result supports our previous observation that the lower sensitivity to vem shown by pigmentable vs. non-pigmentable cells is a consequence of their ability to become pigmented upon vem treatment. Crucially, it also shows that pigmentation fully explains such a difference in sensitivity, with no other biological process prominently involved. Finally, it points toward a new angle of the pigmentation process that we had overlooked: melanosomes need to remain intracellular to exert their protective function against vem [25], while the increase in their number can be due not only to an increase in maturation, but also to an increase in trafficking. In turn, such findings provide the rationale for combining BRAFi with a new class of pigmentation inhibitors. We had already shown that inhibitors of TYR expression (e.g., LNA-211, the inhibitor of miR-211) or activity (e.g., phenylthiourea) cooperate with vem, by “whitening” pigmentable cells, hence rendering them equivalent to non-pigmentable ones [25]. Here, we showed that the same outcome can be obtained using inhibitors of melanosome uptake, such as heparin, while we propose that promoters of melanosome release should behave similarly.

In summary, pigmentation is an early adaptive response that fully accounts for the lower sensitivity to BRAFi displayed by pigmentable vs. non-pigmentable melanoma cells. Since such a response depends on the number of mature melanosomes contained within the cytoplasm of BRAFi-treated cells, inhibitors of melanosome maturation and/or melanosome trafficking should be systematically tested to fully unleash the efficacy of BRAFi on pigmentable and non-pigmentable cells alike.

## Figures and Tables

**Figure 1 cancers-15-00894-f001:**
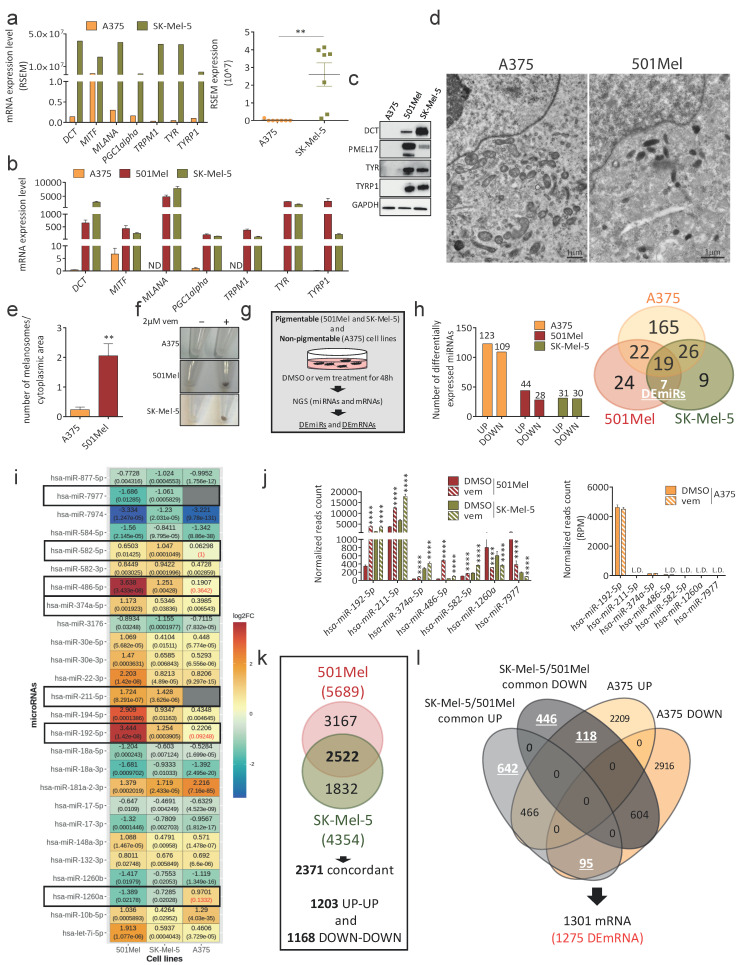
DEmiRs and DEmRNAs in vem-treated pigmentable 501Mel and SK-Mel-5 cells vs. non-pigmentable A375 cells. (**a**–**c**) mRNA level ((**a**), CCLE data (https://sites.broadinstitute.org/ccle/ (accessed on 31 August 2022)): *left*, single genes; *right*, grouped genes); (**b**), qRT-PCR) and protein level ((**c**), Western blot) of genes involved in melanin synthesis, as detected in pigmentable melanoma cells (501Mel (dark red) and SK-Mel-5 (olive green)) and in non-pigmentable A375 melanoma cells (orange). (**d**,**e**) Representative images (**d**) and quantification (**e**) of melanosomes detected by TEM in A375 and 501Mel cells. Scale bar: 1 µm. The graph represents the mean ± SEM of 3 independent experiments. (**f**) Representative images of cell pellets after 72 h of treatment of A375 (*top*), 501Mel (*middle*) and SK-Mel-5 cells (*bottom*) with DMSO or 2 µM vem. Unlike non-pigmentable A375 cells, pigmentable 501Mel and SK-Mel-5 cells are characterized by vem-induced darkening. (**g**) Cartoon of the experimental workflow that has been followed to obtain NGS data. (**h**) (*left*) Number of up- and down-regulated miRNAs in A375 (orange), 501Mel (dark red) and SK-Mel-5 (olive green). (*right*) Venn diagram of cell line-specific and common DE miRNAs in the three cell lines. (**i**) Heatmap of the 26 DE miRNAs shared by 501Mel and SK-Mel-5 pigmentable melanoma cells. Nineteen are shared by A375 cells as well, while seven DEmiRs (highlighted with black rectangles) are not: miR-192-5p, miR-211-5p, miR-374a-5p, miR-486-5p and miR-582-5p are up-regulated in vem compared to DMSO; miR-1260a and miR-7977 are down-regulated in vem compared to DMSO. The adjusted *p*-value is reported below each log2FC value. (**j**) Normalized reads count of miR-192-5p, miR-211-5p, miR-374a-5p, miR-486-5p, miR-582-5p, miR-1260a and miR-7977 DEmiRs, after 48 h of treatment with DMSO or 2 µM vem. Dark red: 501Mel cells; olive green: Sk-Mel-5 cells; orange: A375 cells. (**k**) Venn diagram of cell line-specific and common DE RNAs in 501Mel and SK-Mel-5 cells. (**l**) Venn diagram of RNAs differentially expressed in a concordant or discordant way in 501Mel plus SK-Mel-5 cells (grey ovals) vs. A375 cells (orange ovals). Pigmentable melanoma cell line-specific DEmRNAs are 1275. ** *p* < 0.01, **** *p* < 0.0001.

**Figure 2 cancers-15-00894-f002:**
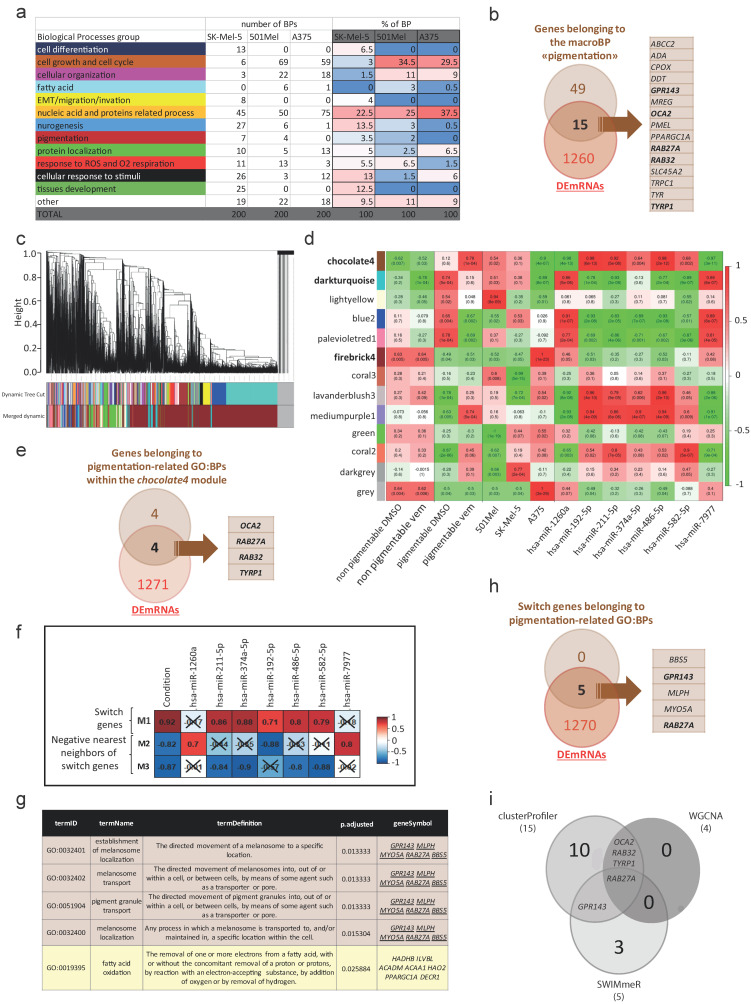
Selective enrichment of the biological process “pigmentation” in pigmentable vs. non-pigmentable cell lines upon vem treatment. (**a**,**b**) clusterProfiler analysis of vem-treated pigmentable 501Mel and SK-Mel-5 cells vs. non-pigmentable A375 cells. (**a**) The 200 top scoring BPs for each cell line mostly fall in the 12 macroBPs listed in the left column. The number and % of BPs belonging to each macroBP are reported in the center and right columns, respectively. The only macroBP that is shared by pigmentable 501Mel and SK-Mel-5 lines and not by the non-pigmentable A375 line is “pigmentation”. (**b**) Venn diagram showing that 15 out of 64 genes belonging to the macroBP “pigmentation” are also DEmRNAs. (**c**–**e**) Weighted gene co-expression network (WGCNA) analysis of vem-treated pigmentable 501Mel and SK-Mel-5 cells vs. non-pigmentable A375 cells. (**c**) Hierarchical cluster tree of 13 modules between the six species (501Mel DMSO and vem; SK-Mel-5 DMSO and vem; A375 DMSO and vem). The branches and color bands represent the assigned module. The color row underneath the dendrogram shows the module assignment determined by the Dynamic Tree Cut algorithm. (**d**) Module trait relationship (*p*-value) for identified modules (y-axis) in relation to traits (x-axis). *chocolate4*, *darkturquoise* and *firebrick4* modules are highlighted in bold. (**e**) Venn diagram showing that 4 out of 8 genes belonging to pigmentation-related GO:BPs within the *chocolate4* module are also DEmRNAs. (**f**–**h**) SWIMmeR analysis of vem-treated pigmentable 501Mel and SK-Mel-5 cells vs. non-pigmentable A375 cells. (**f**) Heatmap of modules-DEmiRs associations. The correlation coefficients between expression profiles of DEmiRs and the modules identified by SWIMmeR (M1, M2, M3) are reported. The color scale ranges from blue (negative correlation) to red (positive correlation). The correlation coefficients that are not statistically significant (*p*-value > 0.05) are marked with an X. (**g**) GO:BPs that are enriched in pigmentable cells-specific switch genes of M1 module and DEmRNAs that comprise them (underlined). (**h**) Venn diagram showing that all 5 switch genes belonging to pigmentation-related GO:BPs are also DEmRNAs. (**i**) Venn diagram showing the 5 DEmRNAs that have been selected by at least two analytical approaches among clusterProfiler, WGCNA and SWIMmeR.

**Figure 3 cancers-15-00894-f003:**
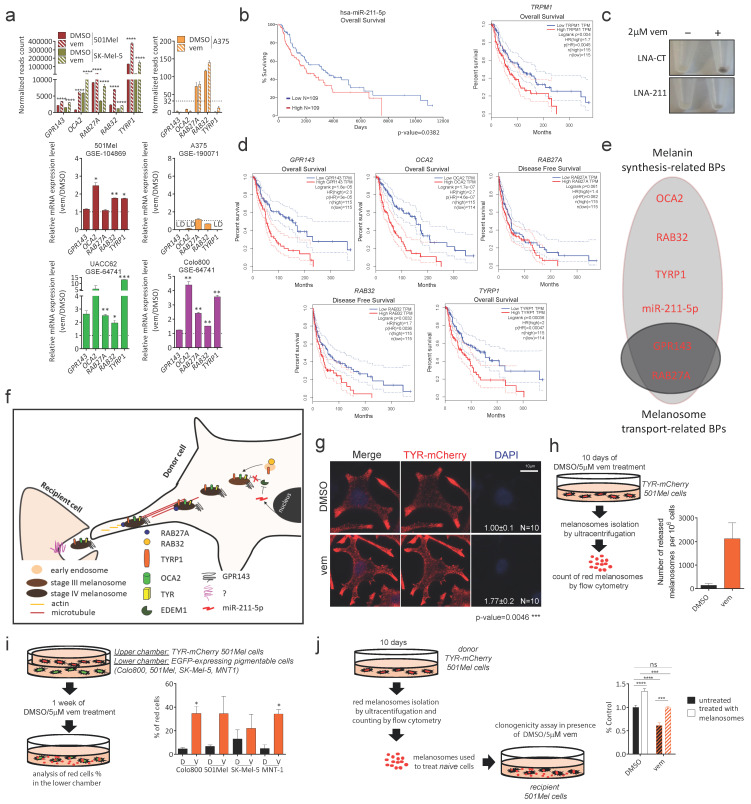
Involvement of DEmiRs and DEmRNAs in melanosome maturation and trafficking upon vem treatment. (**a**) Normalized reads count of *GPR143*, *OCA2*, *RAB27A*, *RAB32* and *TYRP1* DEmRNAs upon vem treatment. (*upper*) Our analysis: 501Mel cells (our RNA-seq, dark red), SK-Mel-5 cells (our RNA-seq, olive green), and A375 cells (GSE89129, orange). (*middle* and *lower*) Additional datasets: 501Mel cells (GSE104869), A375 cells (GSE190071), UACC62 cells (GSE64741), and Colo800 cells (GSE64741). LD: Low detection. (**b**) Kaplan–Meier curves for miR-211-5p *(left*) and its host gene *TRPM1 (right)*, showing significantly worse overall survival for high miR-211-5p/*TRPM1* expressors (25% percentile, red) compared to low miR-211-5p/*TRPM1* expressors (25% percentile, blue). Kaplan–Meier curves were generated using the SKin Cutaneous Melanoma-SKCM dataset on http://www.oncolnc.org/ and http://gepia.cancer-pku.cn/ (accessed on 31 August 2022), respectively. (**c**) Melanin content in 501Mel cells, 96h after the transient transfection with LNA-CT or LNA-211, and 72 h after treatment with 2 µM vem or DMSO vehicle. (**d**) Kaplan–Meier curves for *GPR143*, *OCA2*, *RAB27A*, *RAB32* and *TYRP1*, showing worse overall or disease free survival for high expressors (25% percentile, red) compared to low expressors (25% percentile, blue). The differences are not statistically significant only in the case of *RAB27A*. Kaplan–Meier curves were generated using the SKCM dataset on http://gepia.cancer-pku.cn/ (accessed on 31 August 2022). (**e**) Schematic representation of miR-211-5p DEmiR and DEmRNAs belonging to *Melanin synthesis-related* BPs and *Melanosome transport-related* BPs. (**f**) Cartoon summarising melanosome maturation and trafficking, with special emphasis on the known role played by miR-211 DEmiR and the 5 DEmRNAs (see text for details). (**g**) Representative images of TYR-mCherry 501Mel cells, upon 72 h of treatment with DMSO vehicle (*upper*) or 2 µM vem (*lower*). The increase in melanosomes caused by vem treatment is appreciated as an increase in red fluorescence and can be quantified using ImageJ software (http://rsb.info.nih.gov (accessed on 31 August 2022)). Scale bar: 10 µm. (**h**) Quantification of melanosome release. Red fluorescent melanosomes released in the supernatant of TYR-mCherry 501Mel cells were isolated by ultracentrifugation and counted by flow cytometry, after 10 days of treatment with 5 µM vem or DMSO vehicle. (**i**) Quantification of melanosome uptake. TYR-mCherry 501Mel cells (*upper chamber*) were co-cultured with EGFP-expressing pigmentable cells (Colo800, 501Mel, SK-Mel-5, MNT-1; *lower chamber*), in presence of 5 µM vem or DMSO vehicle. The % of red cells present in the lower chamber was measured by flow cytometry after one week. (**j**) Clonogenicity assay performed on 501Mel cells treated with isolated melanosomes. Red fluorescent melanosomes released by TYR-mCherry 501Mel cells over the course of 10 days were isolated and counted as described in (**h**). They were then used to treat naïve 501Mel cells. The number of colonies counted after 10 days of treatment with 2 µM vem or DMSO vehicle is reported. The graphs in (**h**–**j**) represent the mean ± SEM of 3 independent experiments. * *p* < 0.05, ** *p* < 0.01, *** *p* < 0.001, **** *p* < 0.0001.

**Table 1 cancers-15-00894-t001:** Sequence of qRT-PCR primers and LNAs.

Type	Gene	Sense (5′-3′)	Antisense (5′-3′)	PMID
**qRT-PCR primers**	*ATPA1*	CTCAGATGTGTCCAAGCAAG	GTCAGTGCCCAAGTCAATG	28445987
	*DCT*	CCTTTCTTCCCTCCAGTGAC	AGCCAACAGCACAAAAAGAC	28445987
	*GAPDH*	CGCTCTCTGCTCCTCCTGTT	CCATGGTGTCTGAGCGATGT	28445987
	*MITF*	TGACCGCATTAAAGAACTAGG	GTGCTCCAGTTTCTTCTGTCG	28445987
	*MLANA*	CTCTTACACCACGGCTGAA	AGACTCCCAGGATCACT	28445987
	*PBGD*	TCCAAGCGGAGCCATGTCTG	AGAATCTTGTCCCCTGTGGTGGA	28445987
	*PGC1alpha*	GTCACCACCCAAATCCTTAT	CGGTGTCTGTAGTGGCTTGA	28445987
	*SDHA*	CCACTCGCTATTGCACACC	CACTCCCCATTCTCCATCA	28445987
	*TRPM1*	TGCGAAGGCTGCTGGAAA	CAAGACGATGGACACCACGTTAGG	28445987
	*TYR*	GATGAGTACATGGGAGGTCAGC	GTACTCCTCCAATCGGCTACAG	28445987
	*TYRP1*	GGACCAGCTTTTCTCACAT	GAATCAAAGTTGCTTCTGGA	28445987
**LNAs**	LNA-CT		GTGTAACACGTCTATACGCCCA	28445987
	LNA-211		AGGCGAAGGATGACAAAGGGAA	28445987

## Data Availability

Data are contained within the article.

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
