# Peer review of "A Network of MicroRNAs and mRNAs Involved in Melanosome Maturation and Trafficking Defines the Lower Response of Pigmentable Melanoma Cells to Targeted Therapy"

_cancers, 2023, doi:10.3390/cancers15030894_

Round 1
Reviewer 1 Report
- The language needs to be polished up substantially - I attempted to correct some parts of the manuscript but soon realised that it will be a futile effort as grammatical mistakes and awkward expression are littered throughout the article. This renders the article difficult to be comprehended even for scientists whose native language is English.
- In the introduction section, there was a swarth of write-up describing the results of the manuscript (line 93-121). The authors should focus on describing the background ideas pertaining to the subject matter and only provide succinct description of the results and hypotheses tested at the end of this section.
- From the results section, I can see that this is mainly a candidate mining expedition with over-the-top description on the bioinformatics methods used to streamline gene candidates purportedly involved in drug response of the pigmented melanoma cell lines. The authors should trim down their description and presentation of the three approaches (clusterProfiler, WGCNA and SWIMmeR).
- The functional experiments presented in Figure 3 seem more like an afterthought - microRNA:mRNA interactions are gleaned from microRNA target prediction software and not validated using functional assays such as microRNA pulldown, luciferase assay etc with the melanoma cell lines. And why go that great length to identify the gene candidates, when their expressions are not perturbed and the outcome of such perturbation investigated. Experiments performed in Fig. 3h-j appears to be disjointed from the rest of the manuscript - if the authors had perturbed the expression of at least 1-2 of the identified candidate genes (overexpression and/or knockdown) to see how they will affect the number of melanosomes released/transferred, then this part of the manuscript will be in coherence with the bioinformatics description.
- The conclusion section is too long, it is almost like a discussion section. Why not separate the result/discussion into two sections, while keeping the description of your conclusion brief and succinct?
Overall, there are quite some interesting ideas that has potential to be showcased to the world but the manuscript in its current form is very raw and requires substantial editing in both language and presentation before it can be submitted to any journal.
Reviewer 2 Report
The authors present a study to understand the resistance mechanisms to BRAFi in melanoma cell lines. It is an important study to understand the resistance mechanisms in patients with BRAF-mutated melanoma, and may help in finding an elusive answer. I do not have any additions, except for one suggestion.
The authors may want to address the question of how checkpoint inhibitors may alter the behavior of cell lines in BRAF-mutated melanoma. With respect to the recent data from DREAMSeq and SECOMBIT trials, in the clinical world, most patients will now receive checkpoint inhibitors first and then BRAF/MEKi second. Do the authors have any data or opinions on this matter? Authors can review the following paper to get more clinical insight- https://ascopubs.org/doi/full/10.1200/OP.22.00267
Round 2
Reviewer 1 Report
I can see improvement in the presentation of the manuscript, especially attempts at correcting the language. There are still language issues that require attention, e.g.:
· Line 63-66: “These miRNAs and mRNAs belong to BPs related to pigmentation, specifically melanosome maturation and trafficking. An increase in the number of intracellular melanosomes - by increased maturation or decreased release/increased uptake - confers in fact resistance to vem.”
Suggest amending to: “These miRNAs and mRNAs belong to BPs related to pigmentation, specifically melanosome maturation and trafficking. In fact, an increase in the number of intracellular melanosomes - by due to increased maturation or decreased release/increased uptake, - confers in fact resistance to vem.”
Note: The use of “in fact” in the original sentence is not appropriate.
· Line 66-68: “Conclusion: We demonstrated that the higher resistance to vem displayed by pigmentable vs non-pigmentable melanoma cells is fully accounted for by their ability to increase the number of intracellular mature melanosomes.”
Suggest amending to: “Conclusion: We demonstrated that the higher resistance to vem displayed by pigmentable vs non-pigmentable melanoma cells, as compared to the non-pigmentable cells, may be due to the former’s their ability to increase their number of intracellular mature melanosomes with drug treatment.
· Line 77-82: “In metastatic melanoma, resistance to BRAF inhibitors (BRAFi) and MEK inhibitors (MEKi) remains one of the main drawbacks of targeted therapy. After an extensive search for genetic resistance mechanisms that occur after prolonged drug exposure, in the recent years adaptive mechanisms that confer early drug tolerance have stepped into the spotlight, because, if appropriately targeted, they allow to prevent drug resistance, rather than reverse it”
Suggest amending to: “In metastatic melanoma, resistance to BRAF inhibitors (BRAFi) and MEK inhibitors (MEKi) remains one of the main drawbacks of targeted therapy. While much research effort over the years was focused on understanding the genetic resistance mechanisms that emerge after prolonged drug exposure, there is growing evidence indicating that adaptive mechanisms that confers early drug tolerance may play an important role in the eventual emergence of resistance mechanism. Targeting these adaptive mechanisms may thus be a feasible way to counteract drug resistance in metastatic melanoma.”
These are more language issues that littered throughout the article – will require the authors to scrub their manuscript again to iron out these hiccups that makes the manuscript tedious to go through.
Also, the conclusion section is still way too long! In fact, I appreciate the conclusion delivered in your abstract, which is succinct and delivered the main ideas of your manuscript very well.
